# An Efficient 2D Protocol for Differentiation of iPSCs into Mature Postmitotic Dopaminergic Neurons: Application for Modeling Parkinson’s Disease

**DOI:** 10.3390/ijms24087297

**Published:** 2023-04-14

**Authors:** Olga S. Lebedeva, Elena I. Sharova, Dmitriy A. Grekhnev, Liubov O. Skorodumova, Irina V. Kopylova, Ekaterina M. Vassina, Arina Oshkolova, Iuliia V. Novikova, Alena V. Krisanova, Evgenii I. Olekhnovich, Vladimir A. Vigont, Elena V. Kaznacheyeva, Alexandra N. Bogomazova, Maria A. Lagarkova

**Affiliations:** 1Federal Research and Clinical Center of Physical-Chemical Medicine of Federal Medical Biological Agency, St. Malaya Pirogovskaya, 1a, 119435 Moscow, Russia; 2Center for Precision Genome Editing and Genetic Technologies for Biomedicine, Engelhardt Institute of Molecular Biology, Russian Academy of Sciences, St. Malaya Pirogovskaya, 1a, 119435 Moscow, Russia; 3Institute of Cytology, Russian Academy of Sciences, Tikhoretsky Ave 4, 194064 St. Petersburg, Russia; 4Vavilov Institute of General Genetics, GSP-1, Gubkina St., 3, 119991 Moscow, Russia

**Keywords:** Parkinson’s disease, iPSCs, disease modeling, differentiation protocol, calcium, SOCE currents, high purity of neuron culture

## Abstract

About 15% of patients with parkinsonism have a hereditary form of Parkinson’s disease (PD). Studies on the early stages of PD pathogenesis are challenging due to the lack of relevant models. The most promising ones are models based on dopaminergic neurons (DAns) differentiated from induced pluripotent stem cells (iPSCs) of patients with hereditary forms of PD. This work describes a highly efficient 2D protocol for obtaining DAns from iPSCs. The protocol is rather simple, comparable in efficiency with previously published protocols, and does not require viral vectors. The resulting neurons have a similar transcriptome profile to previously published data for neurons, and have a high level of maturity marker expression. The proportion of sensitive (SOX6+) DAns in the population calculated from the level of gene expression is higher than resistant (CALB+) DAns. Electrophysiological studies of the DAns confirmed their voltage sensitivity and showed that a mutation in the *PARK8* gene is associated with enhanced store-operated calcium entry. The study of high-purity DAns differentiated from the iPSCs of patients with hereditary PD using this differentiation protocol will allow for investigators to combine various research methods, from patch clamp to omics technologies, and maximize information about cell function in normal and pathological conditions.

## 1. Introduction

Parkinson’s disease (PD) is a progressive neurodegenerative disease. PD is caused by the death of dopaminergic neurons (DAns) of the substantia nigra. Current treatments can slow the progression of the disease, but not cure it. PD can be caused by environmental factors, for example, traumatic brain injury [1,2], insecticide poisoning such as 1-methyl-4-phenyl-1,2,3,6-tetrahydropyridine (MPTP) and paraquat [2] and long-term exposure to manganese [3]. Another reason for the development of PD is hereditary factors. Mutations in *PARK2*, *PARK8*, *GBA*, *SNCA*, *PINK1* and others are described as a cause of PD [4,5,6].

Studying the pathogenesis of neurodegenerative diseases (including PD) and screening potential drugs requires an adequate model of the disease. Studies on patients are limited to postmortem brain samples only. The use of model objects is difficult because laboratory animals do not suffer from many neurodegenerative diseases including PD. Moreover, the physiological adequacy of modeling PD in animals is questionable. When simulating this pathology, the damage of a specific type of nerve cells is induced by the introduction of chemical toxins or increased expression of a mutant gene, associated with disease development. A model based on such a non-physiological way of PD induction in a laboratory animal cannot always be comparable to humans in terms of disease symptoms, which complicates the study of the pathogenesis of the disease [7,8]. This problem can be solved by developing a disease model based on induced pluripotent stem cells (iPSCs) derived from patients’ samples carrying PD-associated mutations. IPSCs are an almost endless source of cellular material and can be differentiated into any type of adult cells, including DAns [9,10]. The use of 2D neuronal differentiation protocols allows for the acquisition of cultures for studying the fundamental cellular, metabolic, and electrophysiological mechanisms of neural networks.

In this paper, we propose a highly efficient and reproducible 2D protocol for the differentiation of iPSCs into DAns. The protocol has been tested on iPSCs derived from PD patients with various mutations and from healthy donors, and on iPSCs derived from fibroblasts using lentiviral vectors and Sendai virus-based vectors. Studies have shown that patient-specific neurons obtained from iPSCs at the transcriptomic level do not differ from similar neurons obtained by other research groups. The proportion of sensitive (SOX6+) DAns in the population, as calculated from the level of gene expression, is higher than resistant (CALB+) DAns [11,12]. We also confirmed that obtained DAns are voltage-sensitive and showed that the calcium influx through store-operated calcium channels is significantly enhanced in PD-specific DAns expressing mutant LRRK2 in comparison with WT DAns, which may underlie the pathogenesis of the *PARK8*-associated form of PD.

Thus, the created model system helps study both the individual characteristics of pathology and the traits common to different genotypes. A high percentage of TH-positive cells and the presence of PD-associated mutations make it possible to use such neuron cultures for omics technologies, single-cell RNAseq and the patch clamp technique.

## 2. Results

### 2.1. Development of an Efficient and Reproducible 2D Protocol for iPSC Differentiation into Tyrosine Hydroxylase-Positive Neurons

Due to the insufficient knowledge of Parkin and LRRK2 function mechanisms in both normal and pathological conditions, cells carrying mutations in the genes encoding these proteins are attractive models for studying PD. Furthermore, the relatively wide prevalence of these genes’ mutations in the population makes such models relevant and important for the development and screening of new drugs.

This study included iPSC lines from three male patients with hereditary PD. Two patients had mutations in the *PARK8* gene, and one patient had mutations in both alleles of the *PARK2* gene (Table 1). iPSC lines derived from skin fibroblasts of the healthy donors were also used. For healthy donors and patients with mutations, the iPSC lines were obtained using both lentiviral vectors and Sendai virus-based vectors.

All iPSCs were characterized according to generally accepted standards. Each cell line had a normal karyotype. The lines obtained from PD patients contained corresponding mutations in their genome. According to PCR analysis combined with reverse transcription and immunocytochemical analysis, all iPSC lines expressed the main markers of the pluripotent state at a level comparable to human embryonic stem cells (ESCs). All iPSC lines have demonstrated the ability to form embryoid bodies and subsequent differentiation into derivatives of three germ layers [13,14] (Appendix A). Line IPSFF1S was deposited to human pluripotent stem cell registry with hPSCreg name RCPCMi007-A [15].

**Table 1 ijms-24-07297-t001:** IPSC lines description.

Patient	Sex	Biopsy Age	Mutation	Fibroblasts	iPSCs	Method	Reference
RG	M	60	-	FRG	IPSRG2L	Lenti	[14]
					IPSRG6L	Lenti	[14]
					IPSRG4S	Sev	[14]
FF	F	48	-	FF	IPSFF1S	Sev	[15]
PDL1	M	60	G2019S in LRRK2	FPDL1	IPSPDL1.4L	Lenti	[14]
			(*PARK8* gene)		IPSPDL1.6L	Lenti	[14]
			and N370S GBA		IPSPDL1.6S	Sev	[14]
PDL2	M	60	G2019S in LRRK2	FPDL2	IPSPDL2.15L	Lenti	[14]
			(*PARK8* gene)		IPSPDL2.6S	Sev	[14]
PDP1	M	65	deletion 202–203 AG	FPDP1	IPSPDP1.5L	Lenti	Appendix A [16]
			in the second exon and splicing mutation in 1 intron (IVS1+1G/A) in *PARK2* gene				

We used the protocol by Kriks S. [10] as a reference point to develop our protocol for directed iPSC differentiation into DAns. Using the published protocol, we failed to achieve the claimed differentiation efficiency in 80% of TH-positive neurons. Perhaps the for this reason was that we used culture media and growth factors from other manufacturers. Following the published protocol, iPSC differentiation began after growing the cells in a medium conditioned on mouse embryonic fibroblasts. We set ourselves the task of eliminating the step associated with the use of animal components of undetermined composition. Therefore, before differentiation, iPSCs were grown in a commercially available medium for human pluripotent cells mTeSR1, which required a change in the incubation time with various differentiation factors. The previously published protocol uses eight media of different compositions, while the protocol we developed uses only three different media, which greatly simplifies its application. We also introduced an additional step of reseeding and/or freezing of neuronal progenitors, which made it possible to preserve neuronal progenitors at different stages of differentiation in liquid nitrogen, saving time and expensive reagents.

The differentiation protocol consisted of three steps. Prior to differentiation, human iPSCs were cultured until 90% of visual confluency in the mTeSR1 medium on Matrigel-coated 6-well plates. As the induction of neuronal differentiation was based on the dual inhibition of the SMAD signaling pathway [17], the culture medium at the first stage was replaced with a medium for neuronal differentiation (see Section 4 for step-by-step protocol) which contains 80 ng/mL Noggin, 10 μM SB431542 and 4 μM dorsomorphin. As shown in the experiments on the frog embryo, the inhibitor of the BMP signaling cascade, Noggin, is one of the main inducers of neural plate formation in the Spemann organizer [18]. SB431542 inhibits Lefty/Activin/TGFβ signaling pathways by blocking the phosphorylation of ALK4, ALK5 and ALK7 receptors [19]. The small molecule dorsomorphin is an inhibitor of the BMP pathway [20]. The use of dorsomorphin partially replaces the expensive recombinant protein Noggin and increases differentiation efficiency [21]. This cocktail determines cell differentiation toward the anterior part of the neural tube, from which the anterior parts of the central nervous system will subsequently develop. After 14 days of differentiation, we managed to obtain an almost pure population of early neural progenitors (also known as neuroepithelial stem or progenitor cells) phenotypically corresponding to neural tube cells (for example, [22]). Early neural precursors are small cells that grow densely in several layers and often form three-dimensional structures of neuronal rosettes (Figure 1B). Cells with distinct morphology, such as squamous monolayer epithelial cells or fibroblast-like spread cells, were hardly found in the culture. Due to the high purity of neural progenitors, the stage of mechanical selection of neural rosettes present in many differentiation protocols is here avoided. This laborious and time-consuming process is replaced by a rapid and cheap reseeding of neural progenitors using Versene solution, resulting in a homogeneous cell suspension. Instead of Versene solution, enzymatic methods of cell passage (trypsin or accutase) can also be used. However, in our opinion, the method with Versene solution is less traumatic for cells. It causes no damage to surface proteins by proteolytic enzymes and there is no need to wash cells off the enzyme by several sequential centrifugations. Neural progenitors at this stage, upon further differentiation, can produce all types of neurons and glial cells, so different subtypes of neurons and glial cells can be obtained from one batch of neural progenitors [23,24]. Such precursors easily tolerate freezing and can be stored in liquid nitrogen. The efficiency of obtaining neural precursors is extremely high: from 1 to 2 × 10${}^{6}$ iPSCs after 14 days of the first stage of the protocol, about 10–15 × 10${}^{6}$ neural progenitors can be obtained (Figure 1).

During the embryonic development of the neural tube, a number of events occur that lead to the complication of its cellular composition and spatial organization. After neural tube closure, the active division of its cells begins, and the neural tube becomes multilayered. An uneven division of cells in the anterior part of the neural tube leads to the formation of three cerebral vesicles. The anterior and posterior cerebral vesicles then subdivide into two secondary vesicles, thus making five cerebral vesicles [25]. At this stage, Sonic hedgehog (Shh) and fibroblast growth factor 8 (FGF8) are responsible for the specialization of DAns in the substantia nigra of the midbrain. Shh is secreted by the floor plate and is responsible for the ventralization of midbrain cells, and FGF8 is expressed at the border between the diencephalon and hindbrain and forms a caudo-rostral gradient [26]. In vitro, for the specialization of neuronal progenitors obtained at the differentiation protocol’s first stage, high concentrations of Shh and FGF8 were used at 200 ng/mL and 100 ng/mL, respectively. Shh in the culture medium can be partially replaced by the small molecule purmorphamine [27]. In this case, the concentration of recombinant Shh in the culture medium can be reduced to 100 ng/mL. At the end of this stage, neuronal precursors change morphologically, and a cell body and process are observed. The length of the process does not exceed three cell body diameters (see Figure 1C). Such neuronal progenitors are already progenitors of ventral midbrain DAns and can be subcultured using Versene solution and cryopreserved. The ability to preserve neuronal precursors in liquid nitrogen at different stages of differentiation makes it possible to create a bank of such differentiated derivatives. There is no need to start differentiation from the iPSC stage every time an experiment with adult neurons is set up, which saves time and other resources.

At the last stage of differentiation, cells are cultured in the presence of recombinant BDNF and GDNF, which promote neuronal maturation. Forskolin is also added to the culture medium, which ensures the selective survival of postmitotic cells while suppressing glial cells [28]. Ascorbic acid is used as an antioxidant. At this stage, neuron reseeding becomes impossible. Morphologically, most neurons at this stage are cells with a small soma and one, two or more processes (Figure 1D).

At each stage of differentiation, cells were stained with antibodies to markers of pluripotent stem cells (OCT4), neuronal progenitors (SOX1 and PAX6) and neurons (β-III-tubulin), as well as tyrosine hydroxylase (TH), a marker of DAns (Figure 2A). It was shown that differentiation markers replace each other in accordance with the stage of differentiation. In addition, no OCT4 expressing cells were observed in the population of differentiated derivatives. The absence of residual undifferentiated cells is necessary for the use of iPSC derivatives in experiments on laboratory animals and cell therapy. The lack of OCT4 expression in differentiated cells was confirmed by real-time PCR (Figure 2B). The expression of β-III-tubulin increased during differentiation and reached a maximum at differentiation day 54 (Figure 2C). The expression of TH was not detected in neural progenitors and ventral midbrain neuronal progenitors (differentiation days 14 and 24) and appeared only after culturing neurons in the maturation medium (Figure 2D). As for TH, DAT1 (Figure 2D,E) expression was detected only after culturing the cells in the maturation medium, on differentiation day 38, with a maximum at differentiation day 45, after which there was a tendency to decrease expression. It should be noted that a significant decrease in DAT1 expression (Figure 2E) at differentiation days 52 and 59 was observed in neurons obtained from patient iPSCs (iPSPDP1.5L), but not in neurons from the healthy donor (iPSRG2L). Visually, we did not observe cell death in any neuronal cultures after transferring the cells to the maturation medium. Perhaps the observed effect is caused by the influence of the PD-associated mutation and indirectly indicates the model’s validity.

The developed protocol of neuronal differentiation takes 38 days. The resulting cell culture can be maintained in the maturation medium in the presence of BDNF, GDNF and ascorbic acid for at least another 40 days. Starting from the 34th day of differentiation, the cells were analyzed by various methods to determine the differentiation efficiency. PCR analysis showed that the resulting neurons express a marker characteristic of mature neurons: synaptophysin (Figure 3A), which is localized in the synaptic vesicle membrane in the presynaptic neuron and is involved in the mechanism of synaptic vesicle fusion with the membrane during neurotransmitter release [29,30,31]. PD develops in adulthood and old age, and fully mature postmitotic neurons in the substantia nigra undergo death. Synaptophysin expression most likely indicates the presence of mature neurons with functional synapses in the culture. To study PD’s pathogenesis, it is most interesting to compare mature mutated neurons to ones without mutation in terms of their functional properties. Furthermore, our neuron cultures express a marker of catecholaminergic neurons, as well as dopaminergic in particular for tyrosine hydroxylase (Figure 3A), which catalyzes the conversion of L-tyrosine into L-DOPA and limits the rate of the entire process of dopamine synthesis. β-III-tubulin, which is involved in the formation of the neuronal cytoskeleton but is not expressed in glial cells, was chosen as a common neuronal marker. According to the results of immunocytochemical staining in differentiated culture on day 38 of differentiation, β-III-tubulin is expressed in neuronal cultures at a high level (Figure 3B). Tyrosine hydroxylase expression is also confirmed at the protein level (Figure 3B).

In comparison to the previously published protocol of directed differentiation into DAns [10], the protocol proposed in this paper is 4 days longer (38 days). However, the ability to store neural progenitors means a bank of progenitors from different lines can be created and used as needed. The differentiation of 1–2 $\times 10^{6}$ iPSCs yields 10–15 $\times 10^{6}$ early neural progenitors (after the first stage of differentiation) or 40–60 $\times 10^{6}$ neuronal progenitors committed to DAns (after the second stage of differentiation). If thawed progenitor cells are used for subsequent experiments, the differentiation time will be reduced to 24 days (when using precursors after the first stage of differentiation) or to 14 days (using precursors after the second stage of differentiation). In addition, banking progenitors makes it possible to standardize experiments with adult neurons, since the material for a number of experiments will come from one differentiation of one portion of cells of the initial iPSC line. Neuronal progenitor banking is also a great advantage for cell therapy applications. The above protocol makes it possible to obtain and bank a batch of characterized ventral midbrain neuronal progenitors sufficient for about 10 transplantations [32].

The presented protocol is used as a standard in our laboratory practice. The use of cells from healthy donors and patients with PD, carrying PD-associated mutations, as well as iPSCs obtained using different methods (lentivirus and Sendai virus transduction) for validation, confirms the reproducibility and effectiveness of the developed protocol. No significant cell death was observed during differentiation. The differentiation efficiency did not depend on the presence of mutation, or the method of obtaining iPSCs. Thus, the differentiation protocol can be used to obtain DAns cultures from iPSCs of any origin with consistent efficiency.

### 2.2. Determination of Neuronal Differentiation Efficiency Using Flow Cytometry

The most convenient, simple and easily standardized method of assessing differentiation efficiency is flow cytometry. Using this method, both surface and intracellular marker representation can be evaluated. However, surface markers are the most convenient for analysis because the staining procedure takes only about an hour and allows for work with living cells, which may be useful for some experimental designs.

The cell adhesion molecule N-CAM (also called CD56) is localized on the cytoplasmic membrane. N-CAM is the most convenient neural marker because it is present on neuronal and glial cells, but not expressed on epithelial cells [33]. There is a small probability that epithelial cells may form under the described differentiation conditions since the inhibition of the Lefty/Activin/TGFβ and BMP signaling pathways is also necessary for differentiation into epithelial derivatives, which, like neurons, belong to the ectodermal germ layer. The applied differentiation protocol makes it possible to obtain a culture almost free of epithelial cells. IPSCs from a healthy donor and iPSCs from PD patients differentiate into a neural line with an efficiency of about 90% (Figure 4 upper panel).

The efficiency of neuronal differentiation was also assessed by the presence of the surface marker CD24 (Figure 4 bottom panel). This cell adhesion molecule is expressed on the surface of neurons and neuronal progenitors, but not glial cells [34]. Using this differentiation protocol, the purity of the population is 90% of CD24-positive cells (i.e., neuronal cells) both when differentiating iPSCs from a healthy donor, and when differentiating iPSCs with mutations in the *PARK8* and *PARK2* genes.

To model PD, it is important to obtain a neuronal culture enriched in DAns. The proposed protocol makes it possible to obtain neuronal cultures consisting of 50–90% TH+ cells. (Figure 5A). Significant differences in the differentiation efficiency of iPSC lines obtained from a healthy donor and PD patients into DAns (TH+) were not found (Figure 5B). Among the studied iPSC lines, a lower differentiation efficiency (about 50%) was observed in a line obtained from a healthy donor, which confirms that dopaminergic differentiation is an individual feature of a particular iPSC line.

The high efficiency of differentiation makes it possible to conduct experiments on a population of neurons without worrying about the issue of reliable detection of the survival or mortality of TH+ neurons, which is a prerequisite for the use of cell cultures as test systems for drug screening. The achieved differentiation efficiency is comparable to the efficiency achieved by Kriks S. [10]. Currently, we are not aware of a more efficient protocol for the directed differentiation of iPSCs into TH+ neurons, which does not use cell sorting or genome modifications.

An important proof of the maturity of a neuronal population is the absence of cell proliferation. Only postmitotic neurons can be mature and functional. To prove complete neuronal maturation using the described protocol, the phase of the cell cycle in populations of neurons on the 65th day of differentiation was analyzed by flow cytometry. The vast majority of cells (up to 90%) in all studied lines IPSRG2L (healthy), IPSPDL2.15L (mutation in the *PARK8* gene) and IPSPDP1.5L (mutation in the *PARK2* gene) are in the G1 phase (Figure 6), i.e., do not proliferate, which allows us to conclude that at this stage of differentiation, the cells are mature postmitotic neurons.

### 2.3. Analysis of Neuron Functional Activity

To prove the functional activity of the resulting neuronal cultures, neuron ability to spontaneously release dopamine was evaluated. On the 34th day of differentiation, the culture medium was replaced with a buffer solution (see Section 4) and incubated for 30 minutes, the conditioned buffer solution was collected and the dopamine content was measured using reversed-phase ion-pair chromatography with amperometric detection. All cell lines studied were shown to be capable of spontaneous neurotransmitter release (Table 2). Additional experiments are required to collect more data on this parameter; however, the functional activity (the ability to synthesize dopamine) of neuronal populations differentiated from both PD iPSC lines and healthy iPSC lines is beyond doubt.

To further prove the functional activity of the neurons, their ability to respond to membrane depolarization was tested. The fluorescent imaging experiments showed that the application of 65 μM KCl resulted in a dramatic increase in calcium concentration in cytosol due to calcium influx through voltage-gated calcium channels (VGCC) (Figure 7A). At the same time, no differences were demonstrated between WT and PD-specific cell lines. Moreover, using the patch clamp technique, we registered VGCC and found no significant differences in maximal amplitudes of calcium currents at the potential of 0 mV in WT (IPSRG4S: 1.67 ± 0.33 pA/pF, IPSFF1S: 1.61 ± 0.39 pA/pF) and PD-specific neurons (IPSPDL1.6S: 1.70 ± 0.30 pA/pF, IPSPDL2.6S: 1.34 ± 0.23 pA/pF) (Figure 7B,C).

There was no difference in differentiation efficiency between iPSCs from healthy donors and iPSCs with mutations. The resulting DAns are postmitotic and functionally active (are capable of neurotransmitter release and have electrical excitability). There were also no differences in these parameters between healthy and mutated DAns. Undoubtedly, DAns differentiated from iPSCs cannot fully reproduce age-related changes in the CNS of adults and the elderly. However, the obtained DAns have the minimum necessary characteristics to proceed as a model of DAns in the brains of patients with PD. In conclusion, the obtained results indicate that the developed method for obtaining high-purity DAn population is effective, and these neurons can be used for further studies of the mechanisms of PD pathogenesis both at the level of single cells and at the level of omics.

### 2.4. Quality and Maturity of Neurons as Confirmed by RNA Sequencing Results

To compare some expression features from our differentiated neurons with neurons produced by alternative protocols, additional data were collected. Datasets were collected according to the following criteria: they contain neurons, are differentiated from iPCS and, optionally, their donors were either healthy people or PD patients. Six datasets were found using comparable RNA library construction methods, and one dataset with the Smart-seq library. The latter was excluded from further analysis due to incomparable gene coverage. The dataset characteristics can be found in Table 3. Most datasets are incomplete and contain only one- or two-stage iPSCs, neuronal progenitors or mature neurons. Complete metadata for each sample can be found in Appendix A.

PCA analysis was performed to assess intra- and interphenotype sample similarity. The results show (see Figure 8) that the three phenotype groups (iPSCs, progenitors and mature neurons) are well separated and form distinguished clusters in the space of the first and second components. At the same time, samples of healthy donors and donors with PD behave similarly without clusters. All mature neurons from set51 [36] were located in the area of the neuronal progenitors, as well as some PD neurons from set32 [37]. Perhaps some differences in the expression profile can be explained by the fact that the meaning of the concept of “mature neurons” is somewhat different for different authors. For example, in set51 [36], samples were collected for analysis 10 days after the last reseeding, and in our case, the cultures matured for 1 month. All neurons from our set are located inside the mature neuron cluster among neurons from all sets with neurons. In our analysis, iPSCs originated from only two datasets, and according to their protocols, two clusters have been formed, namely set51 [36] of 2 samples and set64 of 12 samples [38]. The clustering of iPSCs may reflect differences in protocols for generating and culturing cells in different laboratories. In general, all samples are comparable and behave according to the claimed phenotype by PCA.

The expression level for some genes was evaluated for all compared datasets. We assume that the level of some expression signatures will reflect the maturity, dopaminergic potential and vulnerability to PD of the different datasets. Genes *MAP2*, *SYP*, *SYNPO*, *SNAP25*, *VAMP2* and *SYT1* have been chosen as mature neuron markers (Figure 9A) because they play critical roles in synapse function and the neuronal cytoskeleton (for example, [40]. According to [41], genes *TH*, *FOXA2*, *LMX1A*, *LMX1B* and *OTX2* have been chosen as midbrain DAn markers (Figure 9A). Genes *KCNJ6* and *CALB1* were also added to the expression signature as A9 and A10 subtype DAn markers, respectively, (Figure 9B). According to the expression signature, mature neurons were produced by all protocols except set51 [36]. Set32 [37] also demonstrates maturity despite its location on the PCA. The DAn signature is not consistent among the datasets. Only the TH level is high and homogeneous across all mature neurons. A high level of TH, OTX2, FOXA2 and KCNJ6 was found in setloc (our dataset), confirming a high proportion of midbrain DAns of the A9 subtype.

*CALB1* expression is observed at a sufficiently high level in almost all datasets. *CALB1* is a marker of the A10 subtype DAns, which are resistant during PD progression [41]. The underrepresentation of the vulnerable neurons marker may be a sign of the protocol not being suitable for modeling PD. In recent years, a number of studies uncovered markers of cells vulnerable to damage and death in the substantia nigra of patients with PD or in in vitro patients’ iPSCs-derived DAns-based models. Genes *SOX6*, *LMO3*, *ALDH1A*, *AGTR1* and *KCNJ6* [11,12,41] were selected for the evaluation of vulnerable neuron representation in a culture of mature neurons (Figure 10). Genes *TMEM200A* and *CALB1* [12] were selected for the evaluation of resistant subtype DAns (Figure 10). Among markers of vulnerable cell populations, only *AGTR1* expression was low, contrasting with other markers, namely *KCNJ6*, *LMO3* and *SOX6*. *AGTR1* was high only in set51 [36] and set36 [35]. *KCNJ6* expression is rather similar in all datasets. *LMO3* and *SOX6* is highly expressed in DAns from our dataset (setloc), which indicates the presence of a sufficiently large number of sensitive subtype DAns (Figure 10). On the other hand, the expression of resistant subtype DAn markers *TMEM200A* and *CALB1* was also detected at a significant level (Figure 9B and Figure 10).

It follows from the presented data that the proposed differentiation protocol results in a population of DAns consisting of a mixture of “sensitive” and “resistant” neurons. To evaluate the suitability of such a DAn population for modeling PD, we calculate CALB1/TH and SOX6/TH ratios as an indirect assessment of the proportion of resistance and sensitivity cells in the DAns cultures (Figure 11). Only set51 and our set exhibit more sensitivity markers than resistance ones. However, in set51 this is ensured, among other things, by a lower level of TH, not only by a high level of SOX6 and lower maturity of the population (similarity to progenitor cells, as seen in Figure 8).

The RNA seq data confirm the high maturity of neurons in the population and the high purity of the DAn population. The prevalence of the sensitive DAns subtype over the resistant subtype indicates the applicability of neurons obtained according to the proposed protocol for modeling PD.

### 2.5. Store-Operated Calcium Entry Is Enhanced in DAns with a Mutation in the PARK8 Gene

Aberrant calcium signaling has been shown in a number of neurodegenerative pathologies including PD, Huntington’s disease (HD), Alzheimer’s disease, spinocerebellar ataxias, etc. [42,43]. In particular, it has been reported that such a ubiquitous pathway for calcium influx as store-operated calcium (SOC) entry (SOCE) plays an important role in HD pathogenesis [44,45,46,47,48,49]. The studies based on GABA-ergic iPSC-based medium spiny neurons demonstrated a strong enhancement in SOCE in HD-specific neurons in comparison with WT neurons [44,48,49]. Here, we recorded currents through the SOC channels in iPSC-derived DAns to evaluate possible alterations in SOCE in PD neuronal models with a mutation in the *PARK8* gene.

The electrophysiological recordings showed that the steady-state amplitudes of the SOC currents in DAns, modeling PD, at the potential of −80 mV, were equal to 3.32±0.83 pA/pF for IPSPDL1.6S and 3.35±0.61 pA/pF for IPSPDL2.6S, whereas in WT neurons the SOCE amplitudes were 1.96±0.32 pA/pF for IPSRG4S and 1.58±0.29 pA/pF for IPSFF1S (Figure 12A–C). Thus, the SOCE amplitude in any PD-specific DAns was significantly higher than in any WT DAns, leading us to suggest the causative role of the *PARK8* gene mutation in the upregulation of SOC channels. Future studies of the panel of isogenic models with the insertion, deletion and correction of this mutation will support or disprove this hypothesis.

All genes associated with the studied electrophysiological activity exhibit contrasting levels of expression in the control and the PD neurons in the setloc dataset. The expression level of *TRPC1* and *STIM1* is lower in neurons from the donor with PD than from the healthy donor. *ORAI1* and *STIM2* demonstrate similar expression levels for both neuron groups (Figure 13).

## 3. Discussion

About 15% of patients have a family history of PD, while about 5–10% have monogenic inherited forms of the disease. Mutations in 19 genes associated with the development of PD were identified [50]. Mutations in the *PARK8* gene encoding the LRRK2 kinase are the most common cause of both autosomal dominant inherited and sporadic forms of PD [51]. *PARK8* encodes a 2527 amino acid protein that has both kinase and GTPase activity. This protein is expressed in the brain, especially in the hippocampus, striatum, brain stem, cerebellum, midbrain and other organs [50,52]. The most common pathogenic mutation is the G2019S mutation in LRRK2, which is responsible for 2–7% of familial cases of PD and 1% of sporadic cases [51]. Mutant LRRK2 can cause PD due to increased kinase activity [53]. However, information on natural substrates of LRRK2 requires supplementation. The participation of this kinase in the process of neurite outgrowth, movement of synaptic vesicles and the process of autophagy has been demonstrated [54,55]. *Drosophila melanogaster* with a mutation in LRRK2 are hypersensitive to the complex I inhibitor rotenone [56]. Neurons containing the G2019S mutation in LRRK2 exhibited reduced oxygen consumption, increased mitochondrial motility and increased production of reactive oxygen species (ROS) [57,58]. In DAns derived from iPSCs of patients with this mutation, the level of mitochondrial respiration is significantly reduced [59]. In addition, DAns derived from iPSCs from patients with the G2019S mutation show increased sensitivity to mitochondrial and proteasome toxins such as 6-hydroxydopamine, rotenone, and the proteasome inhibitor MG-132 [60]. LRRK2 interacts with parkin (one of the main proteins involved in the utilization of damaged mitochondria) through its ROC (Ras-like G-domain, GTPase) domain. However, this interaction does not lead to increased ubiquitinylation of LRRK2 and probably plays a regulatory role [61]. It was shown that LRRK2 and parkin are part of the same pathway [62].

Here, we also showed that DAns derived from the iPSCs of patients with the G2019S mutation demonstrate significantly higher SOCE amplitudes in comparison with DAns derived from healthy donors’ iPSCs (Figure 12). However, previously published gene expression analysis data identified reduced levels of key store-operated Ca^2+^ entry regulators (STIM1 TRPC1 and ORAI1) in LRRK2 G2019S iPSC-derived neurons compared to their isogenic control [63]. In our data, *STIM1* and especially *TRPC1* demonstrated lower expression in LRRK2 G2019S iPSCs-derived neurons compared to the healthy control, while *ORAI* and *STIM2* did not demonstrate any difference between healthy and PD cell lines. The possible contradiction between enhanced SOCE and reduced mRNA levels of the SOCE machinery components may be resolved by the studies which reported that the G2019S mutation in LRRK2 resulted in overall enhanced translation [64], particularly of multiple genes involved in Ca^2+^ regulation [65]. Thus, reduced mRNA levels may be a compensatory mechanism aimed at the protection of cells from excessive calcium levels. Observed alterations in SOCE, as well as possible changes in ER calcium content, may underlie PD pathogenesis and neuronal death. In particular, ER calcium efflux was shown to be an important regulator of mitochondrial oxidative phosphorylation in pacemaking substantia nigra dopaminergic mice neurons, suggesting its role in neuronal decline with aging and disease [66]. The central role of mitochondrial alterations in neurodegenerative diseases, including PD, is widely discussed [67,68]. In particular, lysosomal mitochondrial clearance was shown to be hampered in PD-specific fibroblasts with the G2019S mutation [63]. Moreover, mitochondrial dysfunction can also be a cause of the elevated activity of the SOC channels. Mitochondrial calcium handling, mitochondrial motility, and ATP production play crucial roles in the formation of high Ca^2+^ microdomains near SOC channels and consequently in calcium-dependent inactivation of ORAI channels [69,70,71]. Future studies should be focused on the coupling of aberrant calcium signaling and mitochondrial dysfunction in neurons, including possible compensatory mechanisms in various neuronal types which may result in selective neuronal vulnerability in different neurodegenerative pathologies.

The increased calcium influx upon membrane depolarization in LRRK2 G2019S iPSC-derived neurons after induction of ER stress by the 24 h thapsigargin treatment has been also demonstrated [63]. At the same time, there were no differences in depolarization-induced calcium influx in LRRK2 G2019S iPSC-derived neurons and its isogenic control without thapsigargin challenge [63], which totally agrees with our data on PD-specific DAns with the G2019S mutation (Figure 12). However, another published paper reported a significant increase in L-type VGCC-mediated calcium currents in iPSC-derived DAns with the G2019S mutation in LRRK2 [65]. Moreover, this elevated calcium influx could be attenuated by the genetic correction of the mutation in LRRK2, indicating the causative role of the mutated LRRK2 in observed alterations of VGCC-mediated calcium entry. Therefore, further studies are needed to clarify whether alterations in calcium influx through VGCC are common features of PD caused by mutated LRRK2 or whether these effects are patient-specific.

Mutations in the *PARK2* gene encoding E3-ubiquitin ligase parkin are the most common cause of juvenile PD inherited by an autosomal recessive mechanism [72]. Mutations in the *PARK2* gene were first identified in five Japanese patients with juvenile PD [73]. Parkin is involved in the utilization of damaged mitochondria [50,74]. Parkin is recruited to the mitochondrial membrane by PINK1 kinase and marks the defective mitochondria, directing it to mitophagy and thereby maintaining mitochondrial homeostasis in the cell. In iPSCs-derived neurons with a mutation in parkin, there is a change in mitochondria morphology under oxidative stress, but mitochondria with normal morphology are also present. When applying CCCP, damaged mitochondria in mutant neurons are not properly utilized [75,76]. Parkin can affect the transport of mitochondria in the cell since it is shown to bind to microtubules [77]. Parkin controls dopamine utilization in DAns of the midbrain by increasing the accuracy of dopaminergic transmission and suppressing dopamine oxidation [78]. Neurons with mutant parkin have an increased level of spontaneous dopamine release due to a decreased number of dopamine transporter binding sites. It has been found that parkin contributes to the degradation of improperly assembled transporters [60].

The above-mentioned cellular dysfunctions are just the tip of the iceberg. Thus, these proteins are attractive for further studies of PD. In addition, the relatively wide population prevalence of mutations in these genes makes the generation of such models relevant and important for the development and screening of new drugs.

Due to the development of cell therapy [32,79,80,81] (NCT02903576; NCT02590692; NCT02286089; NCT03305029), developing approaches for targeted differentiation of iPSCs and ESCs necessitates consideration of the fact that the developed protocol can also be used for mass cell growth in clinical practice. For both clinical and laboratory tasks, protocol cost matters. It is preferable to replace expensive recombinant growth and differentiation factors with cheaper and small molecules. The rate of differentiation is also very important. It is useful to develop a bank of progenitor cells which will save time and will allow for a stock of differentiated derivatives characterized and suitable for transplantation. The simplicity of preparation is of particular importance for the cell product’s clinical use, i.e., the differentiation protocol should not contain steps that require non-standard or highly specialized cell culture skills, as this will limit its widespread use. A necessary requirement is the reproducibility of the differentiation protocol. Before using the differentiated cell population for laboratory and clinical practice, it is necessary to verify the cells’ safety and functionality. There should be no residual undifferentiated cells in the differentiated population that can provoke the formation of teratoma after transplantation [32]. To assess the quality of the resulting cell population, it is necessary to apply appropriate functional tests (for example, electrophysiological activity for neurons, insulin synthesis in response to the addition of glucose for beta cells of the Langerhans islets, albumin synthesis for hepatocytes, etc.).

Another important point for cell therapy is the absence of modifications in the cell genome. For directed differentiation, approaches associated with the delivery of differentiation factors into the genome in the form of integrating vectors can be used. The method based on adding of a vector carrying a gene responsible for directed differentiation into the cell genome is the most effective. Lmx1a is one of the main transcription factors required for the differentiation of DAns in the midbrain. The addition of a vector expressing the Lmx1a gene into the cell genome can increase the efficiency of differentiation of human pluripotent cells into DAns and yield a more homogeneous neuronal population [82]. Another method for increasing the efficiency of iPSC differentiation into DAns is the introduction a vector containing a fluorescent protein under the control of the Lmx1a gene promoter into the genome [83]. At a certain stage of differentiation, this makes it possible to select precursors of DAns using the cell sorter. However, viral integrations into the cell genome can lead to malignant transformations of the cell if any important gene is damaged, and are not approved for use in clinical practice. Thus, differentiated derivatives of these cells are not suitable for cell therapy, meaning the development of a highly efficient differentiation protocol into DAns and its precursors without making modifications to the genome of differentiating cells is crucial.

To study PD in laboratory conditions, the purity of the differentiated population is of great importance, since DAns differentiated from iPSCs of patients with PD are in some cases more sensitive to oxidative stress and autophagy inhibitors than other types of neurons differentiated from the same iPSCs [60]. The determination of the efficiency of neuronal differentiation is necessary for the use of differentiated iPSC derivatives as a model for studying PD. The method for detecting the phenotype of the studied disease and detecting changes in drug screening should be selected based on the percentage of target cell type in the population. Most studies performed on iPSCs with PD-causing mutations did not compare to the efficiency of differentiation into dopaminergic neurons between PD and healthy iPSC [57,62,78,84]. Only one study observed differences in the efficiency of differentiation into DAns between PD and healthy iPSC; however, these differences did not depend on the presence or absence of a mutation in the initial iPSC lineage and, according to the authors, could be explained by individual differences in iPSC lines [82]. However, when studying iPSCs obtained from PD patients with a mutation in the LRRK2, differences were observed in neural progenitors at late passages. Neuronal progenitors carrying the mutation, after prolonged passage, lost their ability to differentiate into MAP2-positive and β-III-tubulin-positive neurons. The morphology of the nucleus had changed, and in such cells the markers of the substantia nigra cells were not found [85]. Thus, under certain conditions, it is possible to detect a pathological phenotype of cells at the progenitor stage of differentiation.

The proposed differentiation protocol meets all of the above requirements. The protocol has the potential to be scaled up by using larger culture vessels. A disadvantage of the protocol is the long period of differentiation—it takes at least 38 days until the characterization of neurons, which in some cases complicates experimental work. However, this time of differentiation is consistent with other published protocols. Another disadvantage of the protocol is the fact that, in order to achieve high differentiation efficiency, a high seeding density (400 thousand/cm${}^{2}$) is required, which may limit studies using microscopy.

According to transcriptome analysis, neurons cultured under our protocol are mature and mainly dopaminergic unlike those of some other protocols and data. Our protocol generates homogenic neurons with a high level of sensitivity markers such as SOX6 and LMO3, which the SOX6/TH and CALB1/TH ratios also confirm.

## 4. Materials and Methods

### 4.1. Reagents and Materials

DMEM (PanEco, Russia)

DMEM/F12 (GIBCO, USA)

mTeSR1 (Stem Cell Technologies, Canada)

FBS (HyClone, USA)

Serum replacement (GIBCO, USA)

Glutamax (GIBCO, USA)

Penicillin-streptomycin (PanEco, Russia)

N2 supplement (GIBCO, USA)

B27 supplement (GIBCO, USA)

Neurobasal (GIBCO, USA)

Neurobasal-A (GIBCO, USA)

GibriS-8 (PanEco, Russia)

### 4.2. Recombinant Proteins and Small Molecule Compounds

Noggin (PeproTech, USA)

Shh (PeproTech, USA)

FGF8 (PeproTech, USA and FRCC PCM own production, Russia)

BDNF (PeproTech, USA and FRCC PCM own production, Russia)

GDNF (PeproTech, USA and FRCC PCM own production, Russia)

Purmorphamine (Stemgent, USA)

Forskolin (Stemgent, USA)

SB431542 (Stemgent, USA)

Dorsomorphin (Stemgent, USA)

ROCK inhibitor (Y27632) (Stemgent, USA)

LDN-193189 (Miltenyi biotec, USA)

Ascorbic acid (Sigma-Aldrich, USA)

### 4.3. Other Reagents for Cell Cultures

Versene solution (PanEco, Russia)

Matrigel (BD Biosciences, USA)

0.05% trypsin solution (HyClone, USA)

Dimethyl sulfoxide (DMSO) (PanEco, Russia)

Trypsin 0.05% (GIBCO, USA)

### 4.4. Molecular Biology Reagents

Phosphate buffered saline (PBS) (PanEco, Russia)

Paraformaldehyde (Sigma-Aldrich, USA)

Tween20 (Sigma-Aldrich, USA)

Goat serum (Hyclone, USA)

Triton X-100 (Sigma-Aldrich, USA)

DAPI (4’,6-diamino-2-phenyliindole dihydrochloride) (Sigma-Aldrich, USA)

RNeasy mini kit (Qiagen, USA)

RNase DNase free set (Qiagen, USA)

Random six nucleotide primers (Sintol, Russia)

MMLV reverse transcriptase (Promega, USA)

Ribonuclease inhibitor (Promega, USA)

Deoxyribonucleotides (Fermentas, USA)

Taq polymerase (Fermentas, USA)

### 4.5. Consumables

Cell culture treated dishes with diameter 35 mm, 60 mm and 100 mm, multi-well plates (Corning, USA), centrifuge tubes, serological pippettes, cryovials (Costar, USA).

### 4.6. Cell Culture Media Compositions

Neural differentiation medium: DMEM/F12, 2% serum replacement, 1% N2 supplement, 1× Glutamax, 50 unit/ml penicillin-streptomycin, 80 ng/mL Noggin, 10 μM SB431542, 2 μM dorsomorphin (instead combination of 10 μM SB431542, 2 μM dorsomorphin and 0.5 μM LDN-193189 can be used).

Neuronal progenitor medium: DMEM/F12, 2% B27 supplement, 1× Glutamax, 50 unit/mL penicillin-streptomycin, 100 ng/mL Shh, 100 ng/mL FGF8 and 2 μM purmorphamine.

NB maturation medium: Neurobasal, 2% B27 supplement, 1× Glutamax, 50 unit/mL penicillin-streptomycin, 20 ng/mL BDNF, 20 ng/mL GDNF, 200 μM ascorbic acid and 4 μM forskolin.

NBA maturation medium: Neurobasal-A, 2% B27 supplement, 1× Glutamax, 50 unit/mL penicillin-streptomycin, 20 ng/mL BDNF, 20 ng/mL GDNF, 200 μM ascorbic acid and 10 μM forskolin.

### 4.7. The Experiment Protocols

#### 4.7.1. IPSCs Cultivation

IPSCs were maintained in mTeSR1 or GibriS-8 media according to the manufacturer’s instructions. Matrigel was used for coating culture plates. Preparation of Matrigel-coated culture dishes and plates was performed according to the manufacturer’s instructions. Cells were passaged using 0.05% trypsin. ROCK inhibitor (Y27632) was added to the culture medium for 1 day after seeding at a concentration of 5 μM. To prepare for cryopreservation, cells were detached using trypsin and washed with DMEM medium containing 10% FBS. After that, cells were resuspended in FBS and transferred to a cryovial. An equal volume of FBS with 20% DMSO was added to the suspension and the cryovial was frozen at −70 ${}^{\circ }$C. In total, 2 million cells were frozen in 1 mL. The next day, cryovials were transferred to liquid nitrogen for long-term storage.

#### 4.7.2. IPSCs Differentiation to Dopaminergic Neurons

IPSCs were detached with trypsin and plated at a density of 40,000 cells/cm${}^{2}$ in mTeSR1 or GibriS-8 medium with presence of 5 μM of ROCK inhibitor.At a density of about 90-100%, the mTeSR1 (GibriS-8) medium was replaced by the neural differentiation medium. Cells were cultured for 14 days with the medium change every other day.The resulting neural progenitors were detached with Versene solution by incubating the cells for 10 min in a CO${}_{2}$ incubator at 37 ${}^{\circ }$C. After that, cells were centrifuged for 5 min at 250× *g*. The cells were washed twice with DMEM/F12 medium, each time centrifuged under the same conditions. Cells were plated at a density of 250,000–400,000 cells/cm${}^{2}$ on cell culture-treated dishes or multi-well plates (depending on the upcoming task) coated with Matrigel and cultured for 10 days in neuronal progenitor medium. The medium was changed every other day.The resulting ventral midbrain neuronal progenitors were detached with Versene solution as described above. Cells were plated at a density of 250,000–400,000 cells/cm${}^{2}$ on cell culture-treated dishes or multi-well plates (depending on the upcoming task) coated with Matrigel and cultured for 7 days in NB maturation medium. The medium was changed every other day. After this stage, the cells could no longer be reseeded.The cells were cultured in NBA maturation medium for at least 1 week until full maturity. For longer culturing, the cells continued to be maintained in the NBA maturation medium with the medium change every other day.

In stages 3 and 4 of this protocol, neuronal progenitors can be subcultured or frozen. Routine passaging was performed with Versene solution as described above. For cryopreservation, the cells were detached and washed, then resuspended in FBS and transferred to cryovials. After that, an equal volume of FB with 20% DMSO was added to the suspension and cryovials were frozen at −70 ${}^{\circ }$C. The next day, cryovials were transferred to liquid nitrogen. In the experiments, the precursors of passages 1–5 were used.

#### 4.7.3. Polymerase Chain Reaction

Total RNA from cell cultures was isolated using the RNeasy mini kit according to the manufacturer’s instructions. DNase treatment was carried out on columns using the RNase DNase free set. RNA concentration was determined using a Qubit (Invitrogen) according to the manufacturer’s instructions.

The reverse transcription reaction was performed using random hexa-nucleotide primers, MMLV reverse transcriptase, ribonuclease inhibitor and deoxyribonucleotides according to the instructions of the reverse transcriptase manufacturer. A total of 1–2 μg of total RNA per reaction was used. To perform PCR amplification, 0.05–0.1 of the reaction mixture from the reverse transcription reaction was used. PCR amplification was performed using Taq polymerase and an Eppendorf thermocycler according to the manufacturer’s instructions (Fermentas). The list of primers used in this work and the amplification conditions are shown in Table 4.

For real-time PCR, a mixture of qPCRmix-HS SYBR + HighROX reagents (Evrogen, Russia) and a CFX96${}^{T}M$ Real-Time PCR Detection thermocycler (Bio Rad, Hercules, CA, USA) were used.

#### 4.7.4. Immunocytochemical Analysis

The cells prepared for staining on cell culture-treated dishes were washed 2 times with PBS, fixed with 4% paraformaldehyde in PBS (pH 6.8) for 20 min at room temperature and washed with PBS with 0.1% Tween 20 3 times. Cell membranes were permeabilized using PBS solution with 0.1% Triton-X100 for 10 minutes at room temperature. Nonspecific sorption of antibodies was blocked by incubation for 30 min in PBS with 0.1% Tween 20 containing 5% FBS and 2% goat serum at room temperature. Antibodies used are shown in Table 5.

Primary antibodies were diluted as recommended by the manufacturer in PBS with 0.1% Tween 20 containing 5% FBS and 2% goat serum, cells were incubated for 1 h at room temperature (surface antigens) or overnight at +4 °C (cytoplasmic and nuclear antigens). After that, cells were washed 3 times for 5 min in PBS with 0.1% Tween 20.

Secondary antibodies (Invitrogen, United States) conjugated with fluorescent labels (Alexa 488, Alexa 555, Alexa 546, Alexa 647) were applied in dilutions recommended by the manufacturer, incubated for 30 min at room temperature in the dark, and washed 3 times for 5 min with PBS containing 0.1% Tween 20. After that, cells were incubated in 0.1 μg/mL DAPI in PBS for 10 min to visualize the cell nucleus, and washed twice with PBS.

The resulting preparations were examined using a Nikon Eclipse Ni microscope.

#### 4.7.5. Flow Cytometry

##### Surface Marker Staining

Mature neurons were carefully detached with Versene solution. Cells were incubated for 10 min in a CO${}_{2}$ incubator and centrifuged for 5 min at 300× *g*. After that, cells were washed twice with DMEM/F12 medium, centrifuged under the same conditions each time.A total of 500,000 cells were resuspended in 100 μL PBS without calcium and magnesium supplemented with 1% FBS. Next, the primary phycoerythrin-labeled anti-N-CAM antibodies (DAKO), the primary FITC-labeled anti-CD24 antibodies (DAKO) and the corresponding isotype control antibodies (DAKO) were added in dilutions recommended by the manufacturer.Cells with antibodies were incubated for 1 h on ice.Cells were centrifuged at 300× *g* for 5 min at +4 °C; the supernatant was removed. Cells were washed once with PBS without calcium and magnesium supplemented with 1% FBS and centrifuged under the same conditions. The cell pellet was resuspended in neuron maturation medium (In total, 100 μL of medium contained 200,000 cells). The cell suspension was mixed with an equal volume of 400 ng/mL DAPI solution and incubated for 5 minutes using ice.Cells were analyzed using an Acea NovoCyte 3000 flow cytometer. A total of 500,000 cells were taken per measurement.The obtained data were analyzed using NovoExpress software. DAPI-negative cells were isolated as a population of living cells; within this population, the proportion of N-CAM-positive and CD24-positive cells was measured.

##### Staining of Cytoplasmic Markers and Staining for Cell Cycle Analysis

Mature neurons were carefully detached with Versene solution. Cells were incubated for 10 min in a CO${}_{2}$ incubator and centrifuged for 5 min at 300× *g*. After that, cells were washed twice with DMEM/F12 medium, centrifuged under the same conditions each time.Cells were resuspended using 1% PFA and incubated on ice for 15 min. Cells were washed twice with PBS without calcium and magnesium supplemented with 1% FBS. After the last wash, 100 μL of washing solution was left and the cell pellet was resuspended in this volume.Next, 80% ethanol was carefully added to the cell suspension. After that, cells were incubated on ice for 30 min.The cells were washed twice with PBS without calcium and magnesium supplemented with 1% FBS.Each sample was resuspended in 200 μL of PBS without calcium and magnesium supplemented with 1% FBS and divided into two tubes. Antibodies against TH (ab112, Abcam) were added to one part, isotype control (026102, Invitrogen) to the other. Samples were incubated at +4 °C overnight.Cells were washed twice using PBS without calcium and magnesium supplemented with 1% FBS. Secondary antibodies conjugated with Alexa 488 (Invitrogen) were added to the cells and incubated on ice at 30 min.The cells were washed twice with PBS without calcium and magnesium supplemented with 1% FBS. The cells were resuspended in the same buffer; in total, 100 μL of solution contained 200,000 cells. A total of 500,000 cells were taken per measurement; an equal volume of 400 ng/mL DAPI solution was added.Cells were analyzed using an Acea NovoCyte 3000 flow cytometer.The rest cells were centrifuged and resuspended in proportion of 200,000 cells/100 μL in 1 mg/mL DAPI solution and PBS with 0.1% Triton-X100. After that, cells were incubated for 30 min on ice. Next, the stained cells were used for cell cycle analysis.Cell cycle analysis was performed by an Acea NovoCyte 3000 flow cytometer using the Automated Cell Cycle Analysis option of NovoExpress Software. The Watson model was used for cell cycle fitting.The obtained data were analyzed using the NovoExpress software.

#### 4.7.6. Measuring Dopamine Production by Neurons

Neurons were cultured in a 24-well plate according to the protocol described above. On the 38th day of differentiation, the cells were washed 3 times with the following buffer: 10 mM HEPES, 114 mM NaCl, 5.3 mM KCl, 1 mM MgCl_2_, 2 mM CaCl_2_, 30 mM glucose, 1 mM glycine and 0.5 mM sodium pyruvate [86]. Cells were then incubated in this buffer for 30 min in a CO${}_{2}$-incubator (37 °C). The conditioned buffer was collected and centrifuged at 250× *g* for 10 min. to remove cellular debris, the supernatant was frozen at −70 °C. The buffer used is similar to the DMEM/F12 medium by its salt composition. This buffer is used for short-term incubation of neuronal cultures, and contains no substances that interfere with the chromatographic determination of dopamine. The dopamine concentration was determined using ion-pair reversed-phase chromatography with amperometric detection (Column–Nucleosil C18 4 × 250 mm). A clean buffer was used as a negative control.

#### 4.7.7. RNA Extraction and Transcriptome Library Construction

One million pelleted cells were used for total RNA isolation with RNeasy Mini Kit (Qiagen, Hilden, Germany) following the manufacturer’s protocol. TURBO DNA-free Kit (Thermo Fisher Scientific, Waltham, MA, USA) was used for DNA traces removal. RNA Integrity Number was determined with BioAnalyzer 2100 instrument (Agilent, Santa Clara, CA, USA) using an Agilent RNA Nano 6000 Kit (Agilent, USA). The average RNA Integrity Number was 8. The initial amount for transcriptome library preparation was 500 ng of total RNA. The NEBNext Poly(A) mRNA Magnetic Isolation Module (New England Biolabs, Ipswich, MA, USA) was used for poly(A) RNA enrichment. Transcriptome libraries were constructed with NEBNext Ultra II Directional Library Prep Kit for Illumina (New England Biolabs, USA) and NEBNext Multiplex Oligos (Dual Index Primers Set 1) for Illumina (New England Biolabs, USA). Paired-end libraries were sequenced on the Illumina NovaSeq 6000 instrument with 2 × 100 cycles (Illumina, San Diego, CA, USA).

#### 4.7.8. Datasets Preparation

Neuron differentiation protocols with neurons, neuron progenitors and iPSCs RNA seq data with fastq and metadata were collected from GEO and SRA archives for comparison with our local data. Quality control by fastQC [87] and MultiQC [88] was performed before and after adapter trimming by cutadapt (version 3.3) [74] and quality filtering by trimmomatic (version 0.39) [89] for the whole dataset. Pseudoalignment was performed by salmon [90] for GRCh38 reference and Gencode.v37 transcriptome. Gene summarizing was performed by tximport [91] using the length-scaled TPM metrics and ignoring the transcripts version. Expression and dispersion for genes and transcripts was estimated by edgeR (version 3.28.1) [92] using the TMM scaling factor and filtering by the minimal total count > 500 through the whole dataset. For further analysis, CPM and RPKM data from edgeR were used. PCA analysis was conducted by PCAtools (version 2.3.9) [93] on log2-transformed CPM removing genes with the lowest 10% variance. Different protocols were also compared using geometric boxplots plotted with ggplot2 (version 3.3.3) [94].

#### 4.7.9. Fluorescent Calcium Imaging

Fluorescent calcium imaging experiments were performed according to previously published protocols [43,44]. Briefly, DAns were loaded with 2 mM Fura-2AM in the presence of 0.025% Pluronic F-127 and incubated in the dark at room temperature for 1 h, then washed by HBSS for 20 min. Cells were illuminated by alternating between 340 and 380 nm excitation light at 2 Hz. The emission fluorescence intensity was measured at 510 nm. The change in cytosolic calcium concentration was expressed as the ratio of emission fluorescence intensities at excitation wavelengths 340 and 380 nm. The voltage-sensitive calcium response was evoked by the application of HBSS supplemented by 65 mM KCl that depolarized the plasma membrane and resulted in the activation of VGCC.

#### 4.7.10. Electrophysiological Recordings

Ion currents were recorded using the whole-cell patch clamp technique as previously described [44,49]. Briefly, to record currents through voltage-gated calcium channels, we maintained cells at −40 mV and applied a series of 500 ms voltage steps, from −80 to +50 mV with an increment of 10 mV. The pipette solution contained (in mM) 125 CsCl, 10 EGTA-Cs, 10 HEPES-Cs, 4.5 CaCl_2_,1.5 MgCl_2_, 4 Mg-ATP and 0.4 Na-GTP; pH was adjusted to 7.3 with CsOH. The extracellular solution contained (in mM) 140 NMDG-Asp, 10 BaCl_2_, 10 HEPES-Cs and 10 Glucose; pH was adjusted to 7.3 with CsOH. The recorded currents were normalized relative to cell capacitance (6–20 pF).

To record currents through SOC channels, the membrane potential was initially held at −40 mV. It was then periodically (every 5 s) decreased to −100 mV for 30 ms, then gradually raised to 100 mV at a rate of 1 mV/ms and then returned to −40 mV. The recorded currents were normalized relative to cell capacitance (5–15 pF). The traces recorded prior to current activation were used as templates for leak subtraction. Currents were evoked by the application of 1 μM thapsigargin to the external solution. The same solutions as for the voltage-gated current recordings were used, with the addition of 0.01 mM nifedipine into the extracellular solution to block the L-type VGCC. All chemicals were obtained from Sigma-Aldrich (USA).

## 5. Conclusions

Ventral midbrain neuronal progenitors obtained according to the proposed protocol can potentially be used for cell therapy, since they do not contain genetic modifications and can be produced in large quantities and frozen for long-term storage. After terminal differentiation, the resultant highly pure, postmitotic, functionally active population of DAns can be successfully used for a wide range of research methods both at the level of the whole population and at the level of individual cells. We demonstrate our protocol’s ability to produce mature DAns with a significant proportion of the sensitive phenotype according to whole transcriptome analysis. We have demonstrated the alterations in calcium signaling in PD-specific neurons with the G2019S mutation in the LRRK2, thus establishing store-operated calcium channels as a promising target for PD treatment.

## Figures and Tables

**Figure 1 ijms-24-07297-f001:**
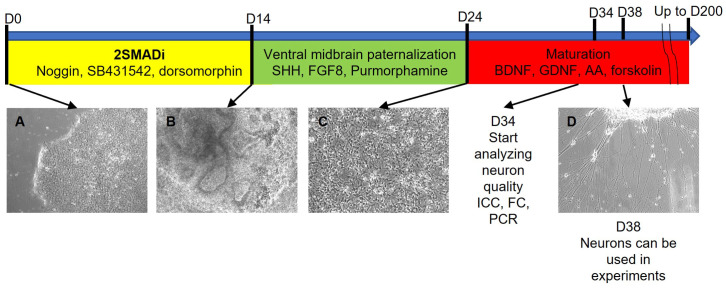
**DAns differentiation protocol scheme.** (**A**)—undifferentiated iPSCs in mTeSR1 media on Matrigel (differentiation day 0); (**B**)—common neural progenitors at the end of the first differentiation step (differentiation day 14); (**C**)—ventral midbrain neuronal progenitors at the end the second differentiation step (differentiation day 24); (**D**)—mature neurons at the third differentiation step (differentiation day 38). Magnification 100×.

**Figure 2 ijms-24-07297-f002:**
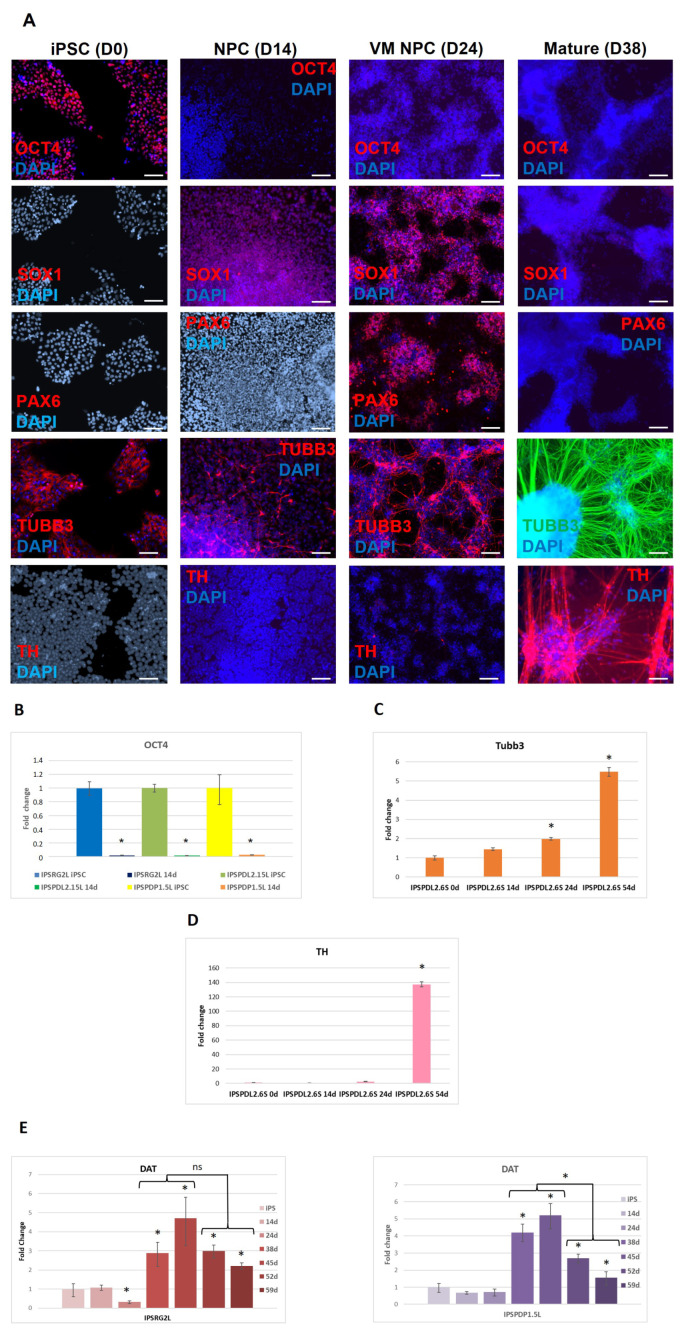
**Analysis of the expression of the pluripotent state marker and neuronal differentiation markers at different stages of differentiation.** (**A**)—immunofluorescence analysis of sequential steps of differentiation for the expression of the pluripotent state marker OCT4, markers of neural progenitors SOX1, PAX6, neuronal marker β-III-tubulin and DAns marker tyrosine hydroxylase (TH) on the example of iPSC line IPSRG2L from a healthy donor. NPC—common neural progenitors (differentiation day 14), VMNPC—ventral midbrain neuronal progenitors (differentiation day 24), Mature—neurons (differentiation day 38). DAPI—blue, corresponding marker green or red. Scale bar 100 μM. (**B**)—Analysis of the expression level of OCT4 by real-time PCR in iPSCs and at differentiation day 14; IPSRG2L—iPSC line from healthy donor; IPSPDL2.15L—iPSC line from PD patient with mutation in *PARK8* gene; IPSPDP1.5L—iPSC line from PD patient with mutation in *PARK2* gene. (**C**)—Analysis of the expression level of β-III-tubulin by real-time PCR in iPSCs and at differentiation days 14, 24, 54 on the example of iPSC line IPSPDL2.6S—iPSC line from PD patient with mutation in *PARK8* gene. (**D**)—Analysis of the expression level of TH by real-time PCR in iPSCs and at differentiation days 14, 24 and 54 on the example of iPSC line IPSPDL2.6S-iPSC from PD patient with mutation in *PARK8* gene. (**E**)—Analysis of the expression level of DAT1 by real-time PCR in iPSCs and at differentiation days 14, 24, 38, 45, 52 and 59; IPSRG2L—iPSC line from healthy donor; IPSPDP1.5L—iPSC line from PD patient with mutation in PARK2 gene. *—the level of expression is statistically significantly different from the level of expression in iPSCs (*p* < 0.05). On the y-axis, the fold changes relative to iPSCs. The bars represent the mean ± SEM.

**Figure 3 ijms-24-07297-f003:**
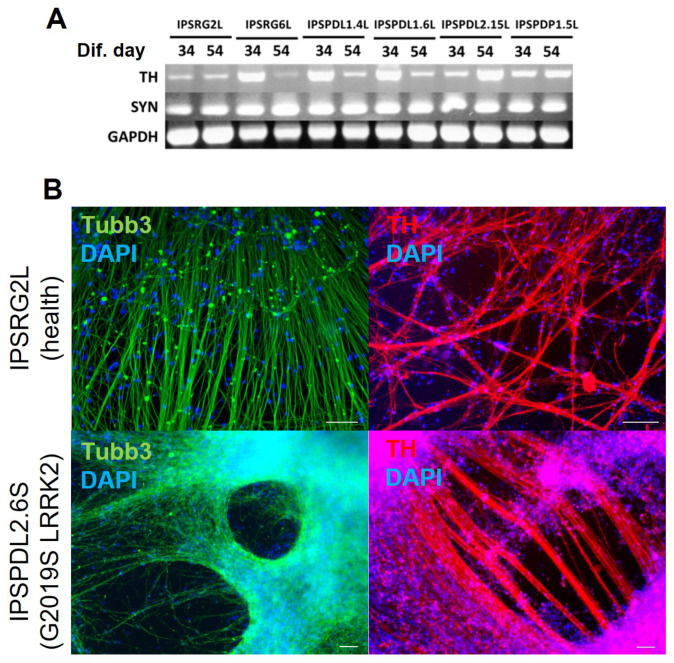
**Analysis of neuronal cultures after 34 days of differentiation for expression of neuronal markers and a marker of the mature state of neurons by PCR and immunocytochemistry.** (**A**)—PCR analysis of neuronal cultures, differentiated from healthy and PD iPSC lines. Expression of common neuronal marker (SYN, synaptophysin) and DAns marker (TH) was shown at 34 and 54 differentiation days. GAPDH—Glyceraldehyde 3-phosphate dehydrogenase. IPSRG2L, IPSRG6L—healthy donor iPSC lines; IPSPDL1.4L, IPSPDL1.6L, IPSPDL2.15L–iPSC lines from two PD patients with mutation in *PARK8* gene; IPSPDP1.5L—iPSC line from PD patient with mutation in *PARK2* gene. (**B**)—immunocytochemical analysis of neuronal cultures differentiated from IPSRG2L (healthy) and IPSPDL2.6S (mutation in *PARK8* gene) IPSC lines on day 45 of differentiation. Green—β-III- tubulin, red—TH, blue—DAPI. Scale bar 100 μM.

**Figure 4 ijms-24-07297-f004:**
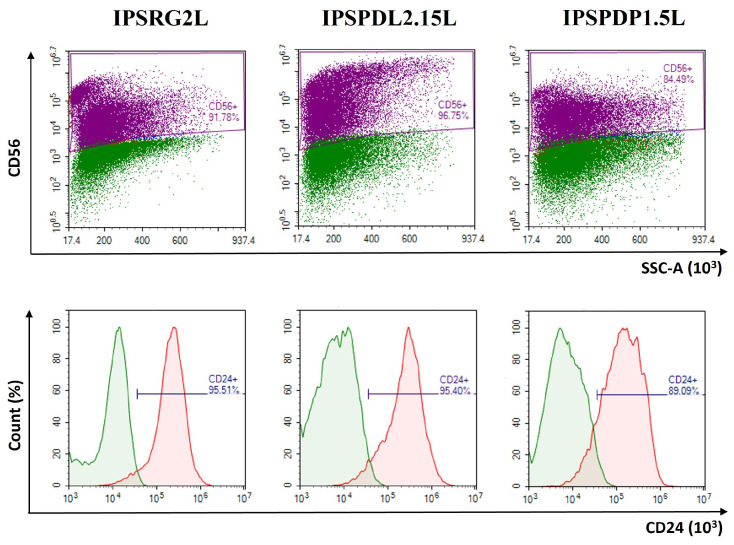
**Results of flow cytometry analysis of neurons differentiated from iPSCs obtained from the material of patients with PD and from a healthy donor, on differentiation day 65. Upper** panel—CD56 (N-CAM)—neural cell adhesion molecule, **bottom** panel—CD24—cell adhesion molecule. IPSRG2L—iPSC line from healthy donor; IPSPDL2.15L—iPSC line from PD patient with mutation in PARK8 gene, IPSPDP1.5L—iPSC line from PD patient with mutation in *PARK2* gene. The data are presented as an overlay image of plots for specific antibody-stained cells and isotype-control antibody-stained cells. Green—isotype control, violet—cells stained for N-CAM, red—cells stained for CD24.

**Figure 5 ijms-24-07297-f005:**
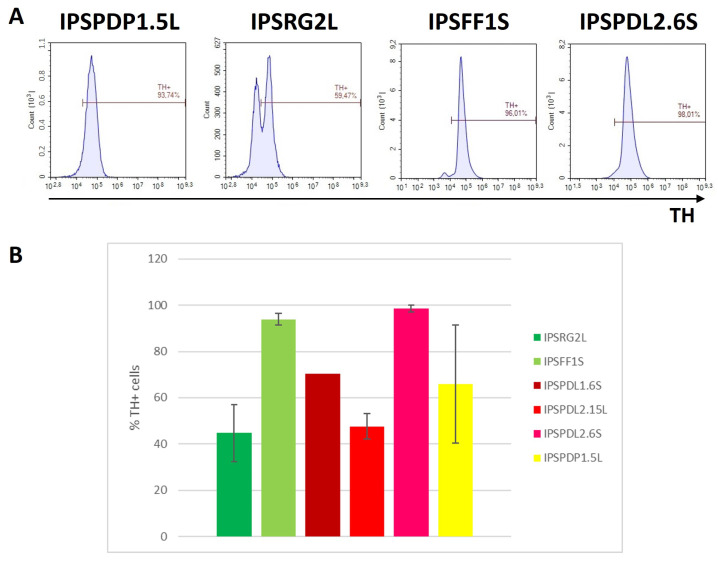
**Flow cytometry analysis of neurons differentiated from iPSCs obtained from the material of patients with PD and from a healthy donor, on differentiation day 65.** (**A**)—anti-TH staining. IPSPDP1.5L—iPSC line from PD patient with mutation in *PARK2* gene; IPSRG2L and IPSFF1S—iPSC lines from healthy donor; IPSPDL2.6S—iPSC line from PD patient with mutation in *PARK8* gene. (**B**)—results of analysis of neuronal populations for TH expression presented in graphical view (n = 1–4). The bars represent the mean ± SEM.

**Figure 6 ijms-24-07297-f006:**
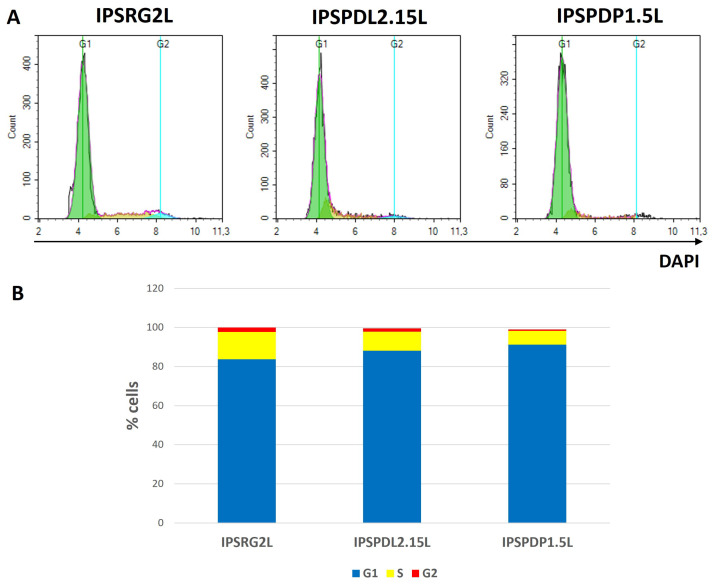
**Cell cycle analysis by flow cytometry.** (**A**)—distribution of cells by phases of the cell cycle for neurons on day 65 of differentiation generated from IPSRG2L line obtained from a healthy donor (left), from IPSPDL2.15L line obtained from PD patient with mutation in *PARK8* gene (middle), from IPSPDP1.5L line obtained from PD patient with mutation in *PARK2* gene (right). (**B**)—the histogram shows the proportion of cells in G1, S and G2 phases of the cell cycle. Statistical data are shown in Appendix A.

**Figure 7 ijms-24-07297-f007:**
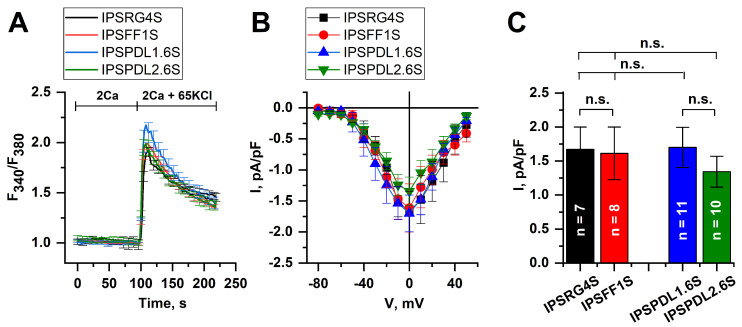
**Depolarization-induced calcium influx in iPSC-derived DAns.** (**A**)—average fluorescence amplitudes of calcium dye Fura-2 AM in DAns derived from PD patients (IPSPDL1.6S, blue line) and (IPSPDL2.6S, green line) with point mutation G2019S in PARK8 gene and healthy donors (IPSRG4S, black line) and (IPSFF1S, red line). The calcium influx was evoked by the application of 65 mM KCl that caused membrane depolarization and consequent calcium entry through voltage-gated channels. The curves are represented as mean ± SEM. (**B**)—average I–V curves of normalized voltage-gated calcium currents in DAns specific for PD patients (IPSPDL1.6S, blue line and IPSPDL2.6S, green line) and healthy donors (IPSRG4S, black line and IPSFF1S, red line). The number of experiments is depicted in panel (**C**). (**C**)—average amplitudes of voltage-gated calcium currents at the potential of 0 mV in DAns specific for PD patients (IPSPDL1.6S, blue bar) and (IPSPDL2.6S, green bar) and healthy donors (IPSRG4S, black bar) and (IPSFF1S, red bar). The amplitudes are represented as mean ± SEM (n = number of single-cell experiments), n.s. indicates the absence of statistically significant differences (*p* > 0.05).

**Figure 8 ijms-24-07297-f008:**
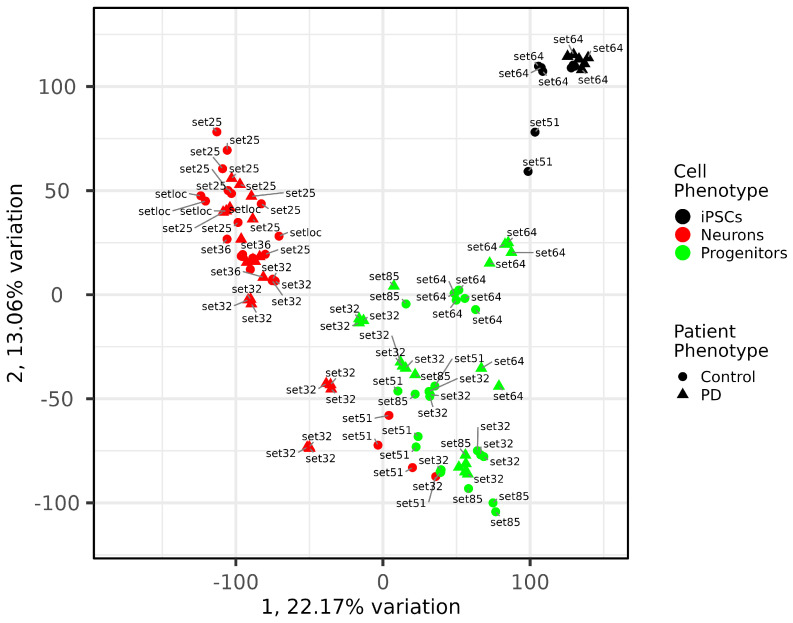
**PCA plot of comparing samples in 7 datasets.** Samples mostly cluster according to their cell type.

**Figure 9 ijms-24-07297-f009:**
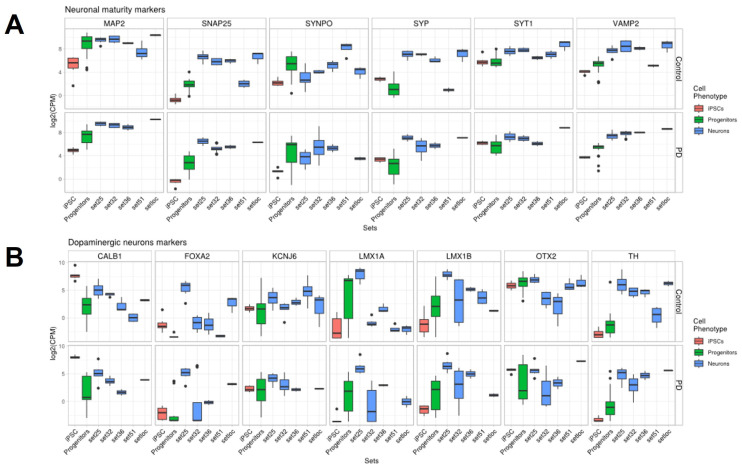
**Expression levels of markers of mature neurons, midbrain DAn as well as A9 and A10 subtypes of DAn in comparison between datasets.** (**A**)—neuronal maturity markers; presents the genes responsible for the functioning of synaptic transmission (*SYP*, *SYNPO*, *SNAP25*, *VAMP2*, *SYT1*) and the cytoskeleton (*MAP2*). (**B**)—midbrain DAn markers (*TH*, *FOXA2*, *LMX1A*, *LMX1B*, *OTX2*), A9 subtype DAn marker (*KCNJ6*) and A10 subtype DAns marker (*CALB1*).

**Figure 10 ijms-24-07297-f010:**
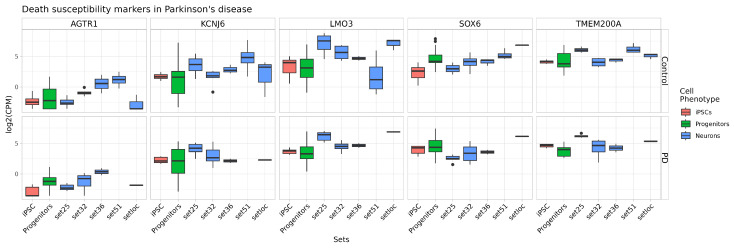
**Expression levels of markers of vulnerable DAn subtype and resistant DAn subtype in comparison between datasets**.

**Figure 11 ijms-24-07297-f011:**
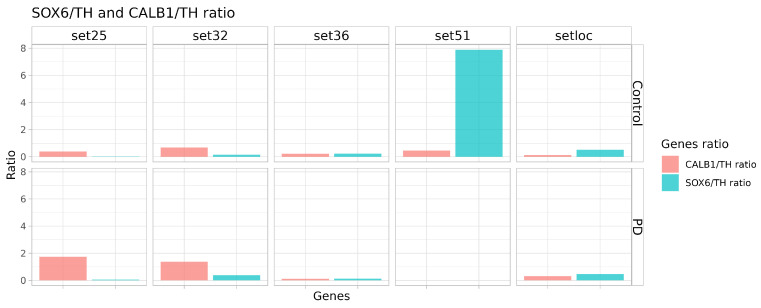
**CALB1/TH and SOX6/TH ratio as an indirect assessment of the proportion of resistant and sensitive cells in the DAn cultures in comparison between datasets**.

**Figure 12 ijms-24-07297-f012:**
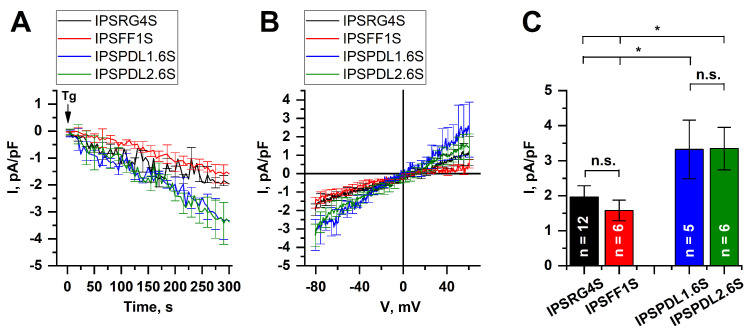
**Store-operated calcium entry in iPSCs-derived dopaminergic neurons.** (**A**)—Normalized SOC currents evoked by application of thapsigargin (1 μM) and represented as a function of time at a test potential of −80 mV in iPSC-based DAns specific to PD patients (IPSPDL1.6S, blue line and IPSPDL2.6S, green line) and healthy donors (IPSRG4S, black line and IPSFF1S, red line). Each trace is represented as mean ± SEM. (**B**)—Average I–V curves of normalized SOC currents evoked by the passive depletion of calcium stores with thapsigargin (1 μM) in iPSC-based DAns specific to PD patients (IPSPDL1.6S, blue line and IPSPDL2.6S, green line) and healthy donors (IPSRG4S, black line and IPSFF1S, red line). The I–V curves were plotted at the steady-state level of the SOC currents. The number of experiments is depicted in panel (**C**). (**C**)—Average amplitudes of the normalized SOC currents at the potential of −80 mV in DAns specific to PD patients (IPSPDL1.6S, blue bar and IPSPDL2.6S, green bar) and healthy donors (IPSRG4S, black bar and IPSFF1S, red bar). The amplitudes are represented as mean ± SEM (n = number of single-cell experiments), n.s. indicates the absence of statistically significant differences (*p* > 0.05). Asterisk indicates statistically significant differences (*p* < 0.05).

**Figure 13 ijms-24-07297-f013:**
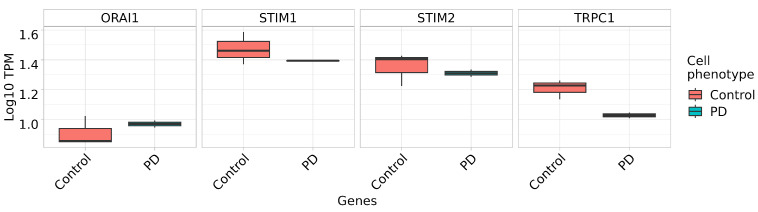
**Expression of genes associated with the electrophysiological activity in setloc dataset.** DAns differentiated from IPSFF1S represent “Control”, DAns differentiated from IPSPDL2.6S (mutation G2019S in LRRK2) represent ”PD”.

**Table 2 ijms-24-07297-t002:** Level of spontaneous dopamine release by TH+ neurons differentiated from iPSC lines derived from PD patients and healthy donor.

Cell Line	pg of Dopamine /10${}^{6}$ TH+ Cells (Mean ± SEM)
IPSRG2L (Healthy)	6.99
IPSRG4S (Healthy)	73.4
IPSFF1S (Healthy)	25.55±1.22 (n = 2)
IPSPDL1.6S (mutation in *PARK8* gene)	51.49 ± 52.28 (n = 3)
IPSPDL2.15L (mutation in *PARK8* gene)	61.21 ± 18.16 (n = 2)
IPSPDL2.6S (mutation in *PARK8* gene)	142.45 ± 63.85 (n = 2)
IPSPDP1.5L (mutation in *PARK2* gene)	72.06 ± 82.04 (n = 2)

**Table 3 ijms-24-07297-t003:** Characteristics of the datasets used.

Author	BioProject	Instrument	PE/SE	Length	Library Type	N	iPSCs	Progenitors	Neurons	Alias
[35]	PRJNA330836	HiSeq 2000	PAIRED	50	polyA	5	0|0	0|0	3|2	set36
[36]	PRJNA699151	NextSeq 500	PAIRED	40	unk	10	2|0	4|0	4|0	set51
	PRJNA761085	NextSeq 500	PAIRED	100	rRNAdepl	10	0|0	5|5	0|0	set85
[37]	PRJNA750432	NextSeq 500	PAIRED	75	polyA	36	0|0	9|9	6|12	set32
[38]	PRJNA767364	HiSeq 1500	SINGLE	50	polyA	24	6|6	6|6	0|0	set64
[39]	PRJNA264625	HiSeq 2500	SINGLE	100	polyA	14	0|0	0|0	8|6	set25
Our data	NA	Novaseq 6000	PAIRED	100	polyA	5	0|0	0|0	3|2	setloc

**Table 4 ijms-24-07297-t004:** Primers used in this study.

Gene	Primer Sequence	T${}_{m}$
TH	5${}^{\prime }$ CCAAGCAGGCAGAGGCCATCATGT 3${}^{\prime }$	60 °C, 30 cycles
	5${}^{\prime }$ GGCGTAGAGGCCCTTCAGCGT 3${}^{\prime }$	
TH (for qPCR)	5${}^{\prime }$ GGGCTGTGTAAGCAGAACG 3${}^{\prime }$	60 °C, 45 cycles
	5${}^{\prime }$ AAGGCCCGAATCTCAGGCT 3${}^{\prime }$	
SYP	5${}^{\prime }$ GCTTTGTGAAGGTGCTGCAA 3${}^{\prime }$	60 °C, 30 cycles
	5${}^{\prime }$ GCCTGAAGGGGTACTCGAAC 3${}^{\prime }$	
GAPDH	5${}^{\prime }$ GAAGGTGAAGGTCGGAGTCA 3${}^{\prime }$	60 °C, 25 cycles
	5${}^{\prime }$ TTCACACCCATGACGAACAT 3${}^{\prime }$	
GAPDH (for qPCR)	5${}^{\prime }$ 5’ GAAGGTGAAGGTCGGAGTCA 3${}^{\prime }$	60 °C, 45 cycles
	5${}^{\prime }$ GTTGAGGTCAATGAAGGGGTC 3${}^{\prime }$	
Oct4 (for qPCR)	5${}^{\prime }$ CAAAGCAGAAACCCTCGTGC 3${}^{\prime }$	60 °C, 45 cycles
	5${}^{\prime }$ TGATCTGCTGCAGTGTGGG 3${}^{\prime }$	
Tubb3 (for qPCR)	5${}^{\prime }$ GGCCAAGGGTCACTACACG 3${}^{\prime }$	60 °C, 45 cycles
	5${}^{\prime }$ GCAGTCGCAGTTTTCACACTC 3${}^{\prime }$	
DAT1 (for qPCR)	5${}^{\prime }$ TGTGGGCTTCACGGTCATC 3${}^{\prime }$	60 °C, 45 cycles
	5${}^{\prime }$ GTCCCAAAAGTGTCGTTGAGG 3${}^{\prime }$	

**Table 5 ijms-24-07297-t005:** Antibodies used in the study for ICC.

Antigen	Dilution	Cat# and Manufacturer
Oct4	1:200	ab18976, Abcam
Sox1	1:500	ab87775, Abcam
Pax6	1:50	ab78545, Abcam
β-III-tubulin	1:1000	ab18207, Abcam
TH	1:2000	ab112, Abcam
Goat-anti-Rabbit IgG	1:800	ab181474, Abcam
Goat-anti-Rabbit IgG	1:800	A32732, Thermo Fisher
Goat anti-Mouse IgG (H+L)	1:800	A11001, Thermo Fisher
Goat anti-Mouse IgG (H+L)	1:800	A21422, Thermo Fisher

## Data Availability

Sequencing data available upon request.

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
