# Peer review of "An Efficient 2D Protocol for Differentiation of iPSCs into Mature Postmitotic Dopaminergic Neurons: Application for Modeling Parkinson’s Disease"

_ijms, 2023, doi:10.3390/ijms24087297_

Round 1
Reviewer 1 Report
The article written by Lebedeva and colleagues descrives a 2D protocol useful for the differentiation of iPSCs into mature postmitotic DA-neurons in feeder-free and integration-free conditions. The authors propose an elegant set of experiments to demonstrate the high purity of the obtained neuronal cell population. However, some minor revisions require to be addressed to increase the reader's understanding.
1. Lane 92: “The use of dorsomorphin partially replace the expensive recombinant protein Noggin and increases differentiation efficiency.” Is this concept a result of the authors or it was previously demonstrated by other studies? It is important to clarify this aspect adding the relative data or the missing reference.
2. References are missing in several parts of the manuscript (in particular in the introduction). Please, add the specific reference when necessary (i.e. lane 96: “Early neural precursors are small cells that grow densely in several layers and often form three-dimensional structures neuronal rosettes”; lane 105: “NPCs at this stage, upon further differentiation can produce all types of neurons and glial cells, so different subtypes of neurons and glial cells can be obtained from one batch of NPCs”; Lanes 110-116: “During the embryonic development of the neural tube, a number of events occur that lead to the complication of its cellular composition and spatial organization. After neural tube closure, active division of its cells begins, and the neural tube becomes multilayered. Uneven division of cells in the anterior part of the neural tube leads to the formation of three cerebral vesicles. The anterior and posterior cerebral vesicles then subdivide into two secondary vesicles, thus making five cerebral vesicles. At this stage, Sonic hedgehog (Shh) and fibroblast growth factor 8 (FGF8) are responsible for the specialization of DAns in the substantia nigra of the midbrain”; and others.
3. Figure 2A: This reviewer suggests to label each panel relative to the immunofluorescence experiments with the name of the target and the color of the fluorophore used for the immunodetection of the antigens. Moreover, It is also suggested to replace the term “immunochemical” with "immunocytochemistry" or "immunofluorescence" experiments in the legend of figure 2. Here, the term mature is also two times repeated (i.e. Mature – mature neurons (differentiation day 38).)
5. Figure 2E: It is not clear to me why the authors labeled the histograms of this graph first with * (that gives statistical significance), and then with a lane (in the upper part of the graph) that indicates the absence of significance (ns). It is suggested to clarify this aspect.
6. Lane 352: “…such as 6-hydroxydopamine, rotenone, and the proteasome inhibitor MG-132 [49]. 2018]”. Please, remove 2018].
7. Lane 560: Material and Methods section “cells were plated at a density of 250,000-400,000 cells/cm2 on Petri dishes or multi-well plates (depending on the upcoming task) coated with Matrigel…”. The authors should clarify if the P100 plates used in this study, that they called Petri dishes, are the dishes usually used for bacterial cultures (and in this case it is correct to define them Petri dishes) or they are 100mm dishes for cell culture, usually defined only P100. This explanation is critical for the reproducibility of the protocol. Usually, Petri dishes are used to induce the “in-suspension” growth of stem cells. Therefore, it is strange that the stem cells plated in Petri dishes are able however to attach and differentiate.
Author Response
Reviewer 1
The article written by Lebedeva and colleagues descrives a 2D protocol useful for the differentiation of iPSCs into mature postmitotic DA-neurons in feeder-free and integration-free conditions. The authors propose an elegant set of experiments to demonstrate the high purity of the obtained neuronal cell population. However, some minor revisions require to be addressed to increase the reader's understanding.
We thank the reviewer for the positive feedback on our work.
- Lane 92: “The use of dorsomorphin partially replace the expensive recombinant protein Noggin and increases differentiation efficiency.” Is this concept a result of the authors or it was previously demonstrated by other studies? It is important to clarify this aspect adding the relative data or the missing reference.
We apologize for missing the citation. Reference to related work was added after specified sentence.
- References are missing in several parts of the manuscript (in particular in the introduction). Please, add the specific reference when necessary (i.e. lane 96: “Early neural precursors are small cells that grow densely in several layers and often form three-dimensional structures neuronal rosettes”; lane 105: “NPCs at this stage, upon further differentiation can produce all types of neurons and glial cells, so different subtypes of neurons and glial cells can be obtained from one batch of NPCs”; Lanes 110-116: “During the embryonic development of the neural tube, a number of events occur that lead to the complication of its cellular composition and spatial organization. After neural tube closure, active division of its cells begins, and the neural tube becomes multilayered. Uneven division of cells in the anterior part of the neural tube leads to the formation of three cerebral vesicles. The anterior and posterior cerebral vesicles then subdivide into two secondary vesicles, thus making five cerebral vesicles. At this stage, Sonic hedgehog (Shh) and fibroblast growth factor 8 (FGF8) are responsible for the specialization of DAns in the substantia nigra of the midbrain”; and others.
References were added in specified places: line 102; line 104, line 114; and in introduction line 37.
Reference [Serruya, M.D., 2017] was deleted (line 34).
Reference [Gilbert S.F., 2003] was replaced to the line 123.
- Figure 2A: This reviewer suggests to label each panel relative to the immunofluorescence experiments with the name of the target and the color of the fluorophore used for the immunodetection of the antigens. Moreover, It is also suggested to replace the term “immunochemical” with "immunocytochemistry" or "immunofluorescence" experiments in the legend of figure 2. Here, the term mature is also two times repeated (i.e. Mature – mature neurons (differentiation day 38).)
Figure 2 modified according to the reviewer’s recommendations. Term “immunochemical” was replaced with "immunofluorescence" in the legend of Figure 2. Repeated term “mature” was deleted (Mature – neurons (differentiation day 38).
- Figure 2E: It is not clear to me why the authors labeled the histograms of this graph first with * (that gives statistical significance), and then with a lane (in the upper part of the graph) that indicates the absence of significance (ns). It is suggested to clarify this aspect.
We thank the reviewer for pointing out to us the insufficient quality of the figure. We modified Figure 2E to clarify statistical significance between groups.
- Lane 352: “…such as 6-hydroxydopamine, rotenone, and the proteasome inhibitor MG-132 [49]. 2018]”. Please, remove 2018].
Sorry for this typo, it has been removed from the text.
- Lane 560: Material and Methods section “cells were plated at a density of 250,000-400,000 cells/cm2 on Petri dishes or multi-well plates (depending on the upcoming task) coated with Matrigel…”. The authors should clarify if the P100 plates used in this study, that they called Petri dishes, are the dishes usually used for bacterial cultures (and in this case it is correct to define them Petri dishes) or they are 100mm dishes for cell culture, usually defined only P100. This explanation is critical for the reproducibility of the protocol. Usually, Petri dishes are used to induce the “in-suspension” growth of stem cells. Therefore, it is strange that the stem cells plated in Petri dishes are able however to attach and differentiate.
We meant standard dishes for adherent cell cultures, we apologize for the misuse of the term. We replaced term “Petri dishes” with “cell culture treated dishes” (Line 545, line 576, line 580, line 608).
Reviewer 2 Report
Dear Authors,
first of all – congratulations for your great effort and interesting topic of the paper. I hope that my hints will be helpful and your final paper will be of great value to the scientific community.
- Despite the well-rooted habit of using abbreviations in Medicine sometimes too many such shortcuts make it more difficult to understand instead of facilitating the topic. It might be worth considering to only use well-known shortcuts, but not to create too many new ones.
- In some paragraphs formatting should be improved, as well as double spaces etc. also, punctation, typos.
- Genes should be written in italics, in order to distinguish them from proteins. In many places of the text it seems to be mixed, therefore difficult to understand properly.
Abstr.line 7: "previously published neurons," --> do we publish neurons...?
Lines 20-21 - not enough, please give examples, elaborate on this topics.
Figure 1 - I guess D0-D14 etc. means day? Not clear.
Figure 2 - maybe it will be good to split this into 2 separate parts, first one consisting the photographs (they're really beautiful) and the second part only the graphs with proper description and bigger letters - now they're very small, thus difficult to understand.
Finally, very good discussion part, well done.
Author Response
Reviewer 2
first of all – congratulations for your great effort and interesting topic of the paper. I hope that my hints will be helpful and your final paper will be of great value to the scientific community.
We are pleased with the positive assessment and the fact that our work has brought a good result. We have tried to improve our article according to your recommendations.
- Despite the well-rooted habit of using abbreviations in Medicine sometimes too many such shortcuts make it more difficult to understand instead of facilitating the topic. It might be worth considering to only use well-known shortcuts, but not to create too many new ones.
We removed the abbreviations NPC and VMNPC from the text.
- In some paragraphs formatting should be improved, as well as double spaces etc. also, punctation, typos.
Thank you for your comment, we have tried to fix formatting errors and typos in the text of the article. All changes are noted in the patch file.
- Genes should be written in italics, in order to distinguish them from proteins. In many places of the text it seems to be mixed, therefore difficult to understand properly.
We checked the names of genes and proteins, the names of the genes are now in italics.
Abstr.line 7: "previously published neurons," --> do we publish neurons...?
We replaced "previously published neurons," with “previously published data for neurons”.
Lines 20-21 - not enough, please give examples, elaborate on this topics.
Thank you for your advice. Information on this topic has been added to the introduction.
Figure 1 - I guess D0-D14 etc. means day? Not clear.
Yes, D0-D14 etc. means differentiation day. We clarify this in the legend of Figure 1.
Figure 2 - maybe it will be good to split this into 2 separate parts, first one consisting the photographs (they're really beautiful) and the second part only the graphs with proper description and bigger letters - now they're very small, thus difficult to understand.
Thank you for your suggestion. We have modified figure 2, enlarged the images B, C, D and E and repositioned the images in the figure.
Finally, very good discussion part, well done.
Thank you very much. We are glad that you liked the article. We hope that our work will be useful.